# Identifying hadronic charmonium decays in hadron colliders

**Nicolo de Groot[1*] and Sergi Castells[2]**

**1** Institute for Mathematics, Astrophysics and Particle Physics (IMAPP),
Radboud University Nijmegen, Heyendaalseweg 135, 6525 AJ Nijmegen, The Netherlands
**2** Department of Physics, University of Illinois at Urbana-Champaign,
1110 West Green Street, Urbana, IL 61801, USA

* N.deGroot@science.ru.nl

## Abstract

Identification of charmonium states at hadron colliders has mostly been limited to leptonic decays of the $J/\psi$. In this paper we present an algorithm to identify hadronic decays of charmonium states ($J/\psi$, $\psi(2S)$, $\chi_{c0,c1,c2}$) which make up the large majority of all decays. The algorithm is able to identify hadronic $J/\psi$ decays with an efficiency of 36% while suppressing a background of quark and gluon jets by a factor 100.

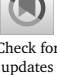

## 1 Introduction

Charmonium states, in particular $J/\psi$, are used both as an analysis and a calibration tool in hadron colliders. The decay to two muons allows for an efficient identification of this state and its narrow peak in the invariant mass spectrum of the muons is a powerful probe of the

momentum resolution of the detector. Only 6% of the $J/\psi$ states decay to a clean final state of $\mu^+\mu^-$ with another 6% decaying to $e^+e^-$. The remaining 88% is decaying to hadronic states and are usually ignored. The other charmonium states are even harder to identify with a small fraction having a clean decay like $\psi(2S) \to \mu^+\mu^-$.

For some analyses it is desirable to obtain as many events as possible, like the search for the rare decay of the Higgs boson to charmonium and a photon, with a branching ratio of around $3 \times 10^{-6}$ for $H \to J/\psi\gamma$ [1,2]. In recent searches by the ATLAS and CMS collaborations [3,4] only the leptonic decays of the $J/\psi$ are used. In this paper we present a tagging algorithm for hadronically decaying charmonium states to obtain higher statistics for these rare decay measurements.

Looking at hadronic decays a couple of things can be noted. The process $H \to J/\psi\gamma$ does not involve QCD fragmentation and further hadronization like in quark and gluon jets. The invariant mass of the final state should be the $J/\psi$ mass, which is lower than the average mass of quark and gluon jets and the average multiplicity is also lower. This is similar to hadronic tau jets, although with a higher mass and multiplicity. Basically we are looking for a fat neutral tau-like object and that is the starting point of our algorithm.

This paper is organized as follows. Section 2 describes our simulation set-up. The observables used to distinguish hadronic charmonium jets are reviewed in Section 3. The neural network architecture and parameters are discussed in Section 4. In section 5 we present our results and we investigate the performance of the network on different charmonium samples and the overall stability of the network.

## 2  Simulated samples

We simulate proton-proton collisions at 13 TeV with the Pythia 8 [5] program. Samples for the 1S charmonium states ($J/\psi$ and $\psi(2S)$) are generated using the gluon-gluon to color-singlet $c\bar{c}$ plus photon or gluon processes (gg2ccbar(3S1) gm/g ). For the 1P charmonium states ($\chi_{c0,1,2}$) the process with the photon is not available and we use the (gg2ccbar(3PJ)g, qg2ccbar(3PJ)q and qq2ccbar(3PJ)g processes which produce an additional gluon or quark. We have verified on the 1S samples that the distributions of the relevant charmonium variables are the same when the $c\bar{c}$ state is produced with a photon or a gluon. In Pythia 8 the minimum transverse momentum in the hard process is set to 30 GeV and the invariant mass of the hard process to within 2.5 GeV of the $Z^0$ mass. This produces jets with very similar momentum distributions as quark jets from $Z^0$ decays. Background gluon jets are taken from the charmonium samples, since they have very similar kinematics as the signal charmonium jets. Background quark jets are taken from a sample of simulated $Z^0$ decays. Long lived particles are allowed to decay if $c\tau < 100$ cm. All samples are simulated without additional pile-up interactions except the sample used to estimate the effects of pile-up.

The events are passed through the DELPHES [6] fast detector simulation using the ATLAS detector configuration files where we use particle-flow jets clustered using the anti-$k_t$ algorithm [7] with a distance parameter $R = 0.4$. The b-tagging settings correspond to an average efficiency for b-jets of 70% with a rejection rate for charm jets of 8.1 and for light quark jets of 440. Jets are said to be charmonium, quark or gluon if the angular distance to a truth charmonium meson, quark or gluon is $\Delta R < 0.2$. Our entire configuration can be found here [8].

# 3 Observables

Since we expect charmonium jets to be similar to hadronically decaying taus, we start off with variables used in the identification of hadronic tau decays by the ATLAS experiment [9]. These variables are using the fact that tau (and charmonium) jets have a lower mass ($m_j$ and $m_{tr}$) and a lower multiplicity ($n_{ch}$ and $n_0$) than quark and gluon jets, are narrower ($\Delta_\eta$, $\Delta_\phi$, $R_{em}$, $R_{track}$) and are not surrounded by further hadronic activity from fragmentation ($p_{core1,2}$, $f_{core1,2}$). To these variables we add the absolute values of the total charge and the jet-charge ($p_T$ weighted charge sum [10]), which are expected to peak at zero for charmonium and gluon jets, but to have a higher average value for jets originating from quarks. Using the output of the b-jet identification algorithm provides some discrimination against b-jets, since the lifetime of charmonium mesons is too short to produce a measurable decay length.

This list of variables is completed with a particular class of generalized angularities [11], which have demonstrated to be efficient in distinguishing quark jets from gluon jets. The angularities depend on two parameters ($\kappa, \beta$) and are defined as:

$$\lambda_\beta^\kappa = \sum_i z_i^\kappa \theta_i^\beta, \tag{1}$$

where $z_i$ is the momentum fraction of jet constituent $i$, and $\theta_i$ is the normalized rapidity-azimuth angle with respect to the jet axis.

Table 1: Input variables to the charmonium classifier

| Name | Description |
|------|-------------|
| $\Delta\eta$ | width of the jet in $\eta$ |
| $\Delta\phi$ | width of the jet in $\phi$ |
| $m_{tr}$ | invariant mass of all charged tracks in the jet |
| $m_j$ | invariant mass of all constituents of the jet |
| $n_{ch}$ | charged particle multiplicity |
| $n_0$ | neutral particle multiplicity |
| abs($Q$) | absolute value of the total charge |
| abs($q_j$) | jet charge ($p_T$ weighted charge sum, $\sum_i q_i \cdot p_{Ti}^{1/2} / \sum_i p_{Ti}^{1/2}$) |
| btag | output of b-tagging algorithm: 1 = b-tagged jet, 0 = not b-tagged |
| $R_{em}$ | Average $\Delta R$ with respect to the jet axis weighted by electromagnetic energy: $\sum_i \Delta R_i \cdot E_i^{em} / \sum_i E_i^{em}$ |
| $R_{track}$ | $p_T$ weighted average $\Delta R$ for tracks: $\sum_i \Delta R_i \cdot p_{Ti} / \sum_i p_{Ti}$ |
| $f_{em}$ | fraction of EM energy over total neutral energy of the jet |
| $p_{core1}$ | ratio of sum $p_T$ in a cone of $\Delta R < 0.1$ and the jet $p_T$ |
| $p_{core2}$ | ratio of sum $p_T$ in a cone of $\Delta R < 0.2$ and the jet $p_T$ |
| $f_{core1}$ | ratio of sum $E_T$ in a cone of $\Delta R < 0.1$ and the jet total $E_T$ |
| $f_{core2}$ | ratio of sum $E_T$ in a cone of $\Delta R < 0.2$ and the jet total $E_T$ |
| $(p_T^D)^2$ | $\lambda_0^2$ with $\lambda_\beta^\kappa = \sum_i z_i^\kappa \theta_i^\beta$ ; $z_i = p_{Ti} / \sum_j p_{Tj}$; $\theta_i = \Delta R_i / R$ |
| LHA | Les Houches Angularity; $\lambda_{0.5}^1$ |
| Width | $\lambda_1^1$ |
| Mass | $\lambda_2^1$ |

The variables are summarized in Table 1. Fig 1 shows the distribution of the variables for $J/\psi$ signal data and a background sample composed of 50% quark jets and 50% gluon jets.

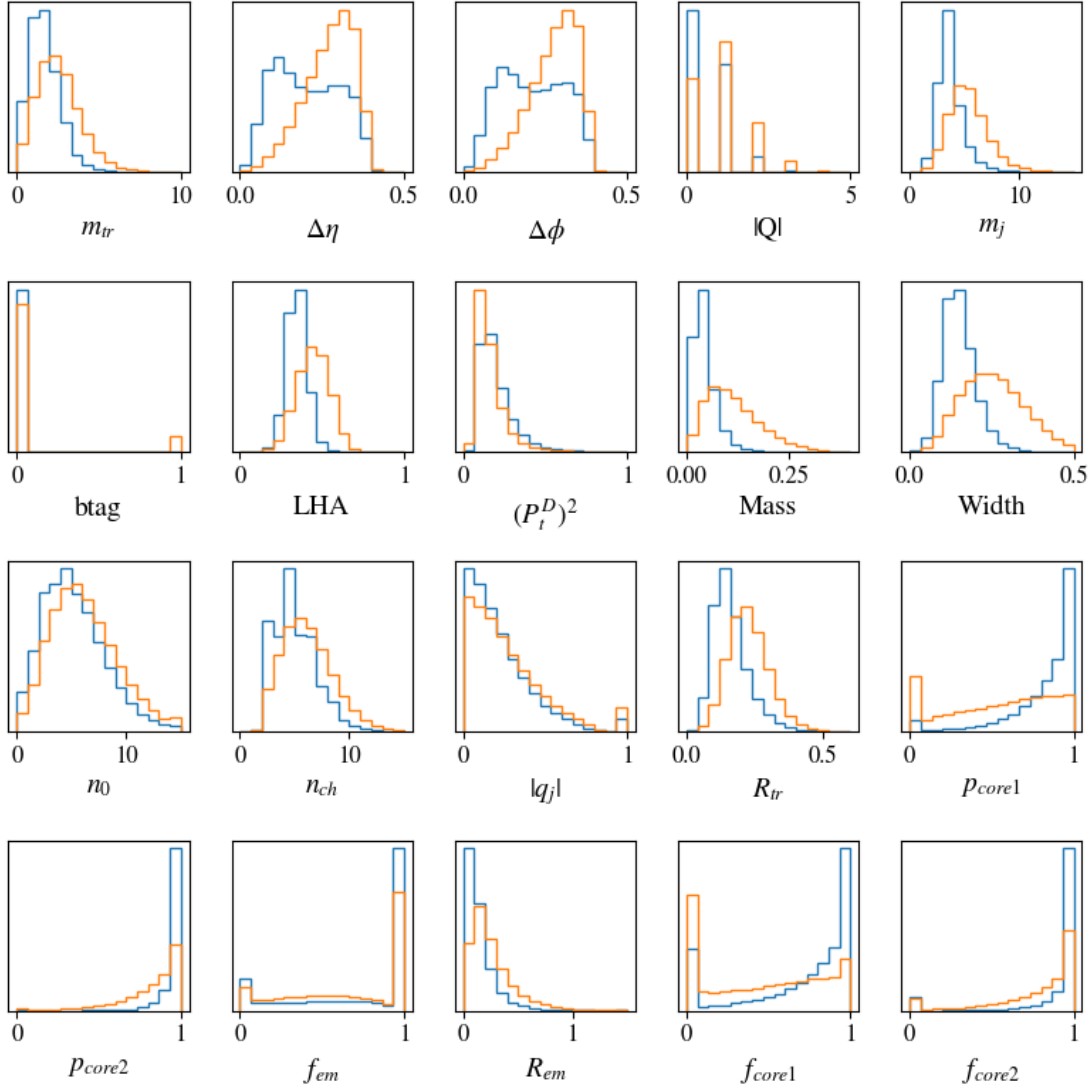

Figure 1: Distributions of the variables used for charmonium identification. Signal ($J/\psi$) in blue, background of a mixed sample of 50% quark and 50% gluon jets in orange. The distributions are normalized to unity.

# 4 Machine Learning

We feed the classifying variables to a fully connected deep network using the TensorFlow [12] and Keras [13] libraries. The network architecture and parameters are listed in table 2. We used the standard technique of dropout layers [14] to prevent overtraining. The network architecture and hyperparameters were optimized by hand on earlier simulation samples. The network performance turns out to be rather insensitive to their value. We use the receiver operating characteristic (ROC) curve as a measure of the separation power of our classifier and in particular the area under the ROC curve (AuC).

For the training we use a sample of 59k simulated hadronic $J/\psi$ decays, 88k quark jets from $Z^0 \rightarrow q\bar{q}$ events and 54k gluon jets. To measure the performance we use independent samples with an equal amount of $J/\psi$, $Z^0 \rightarrow q\bar{q}$ and gluon jets.

Table 2: Hyperparameters of the neural network

| Parameter | Value |
|---|---|
| # layers | 5 |
| Architecture | 20 - 16- 12 - 10 -1 |
| Loss function | binary cross-entropy |
| Learning rate | 0.03 |
| Activation | relu, layer 5: sigmoid |
| Optimizer | Adam |
| Dropout rate | 0.2 |
| Training epochs | 16 |
| Batch size | 64 |

## 5 Results

Figure 2 shows the output distribution for our classifier for both signal and background and the ROC curves. A good separation can be observed with an area under the curve of 0.930. This corresponds to a signal efficiency of 36(15)% at a background rejection factor of 100(1000). It should be noted that the performance of the classifier is significantly better against a background of gluon jets only (AuC = 0.966, signal efficiency of 55(23)% at 100(1000) times background rejection) than against quarks jets (AuC = 0.894, efficiency of 28(11)% at factor 100(1000) background rejection). For gluons the performance against a single background can be further improved to an AuC of 0.975 by using a gluon only background training sample.

Table 3: Overview of the training results. Mixed background test samples contain 50% quark and 50% gluon jets

| Test | Training sample | AuC |
|---|---|---|
| $J/\psi$ vs mixed | $J/\psi$ vs mixed | 0.930 |
| $J/\psi$ vs gluon | $J/\psi$ vs mixed | 0.966 |
| $J/\psi$ vs quark | $J/\psi$ vs mixed | 0.894 |
| $J/\psi$ vs gluon | $J/\psi$ vs gluon | 0.975 |
| $J/\psi$ vs quark | $J/\psi$ vs quark | 0.897 |
| $\psi(2S)$ vs mixed | $J/\psi$ vs mixed | 0.894 |
| $\chi_{c0,1,2}$ vs mixed | $J/\psi$ vs mixed | 0.914 |
| $\psi(2S)$ vs mixed | all $c\bar{c}$ vs mixed | 0.900 |
| $\chi_{c0,1,2}$ vs mixed | all $c\bar{c}$ vs mixed | 0.917 |

The network performs well on jets from the hadronic decay of other charmonium states like the $\psi(2S)$ and $\chi_c$ states, with the area under the curve only a few percent lower than for $J/\psi$. Some of this difference can be recovered by including the heavier charmonium states in the training sample. An overview of the performance of the network for various training and test sets is given in table 3.

We evaluate the network performance dependence on the jet transverse momentum. After reconstruction the transverse momentum spectra of the jets is not completely identical. Gluon jets of around 40 GeV often extend beyond a $\Delta R$ of 0.4 and not all objects are included in its energy measurement. For jets from $b$ and $c$ quarks the neutrinos in semileptonic decays cause a low energy tail to the jet energy spectrum. Some of the variables we use, like the width, correlate with the jet energy. We verify that the network performance does not depend strongly on jet energy by splitting the test sample in three bins: $p_T < 35$ GeV; $35 < p_T < 50$

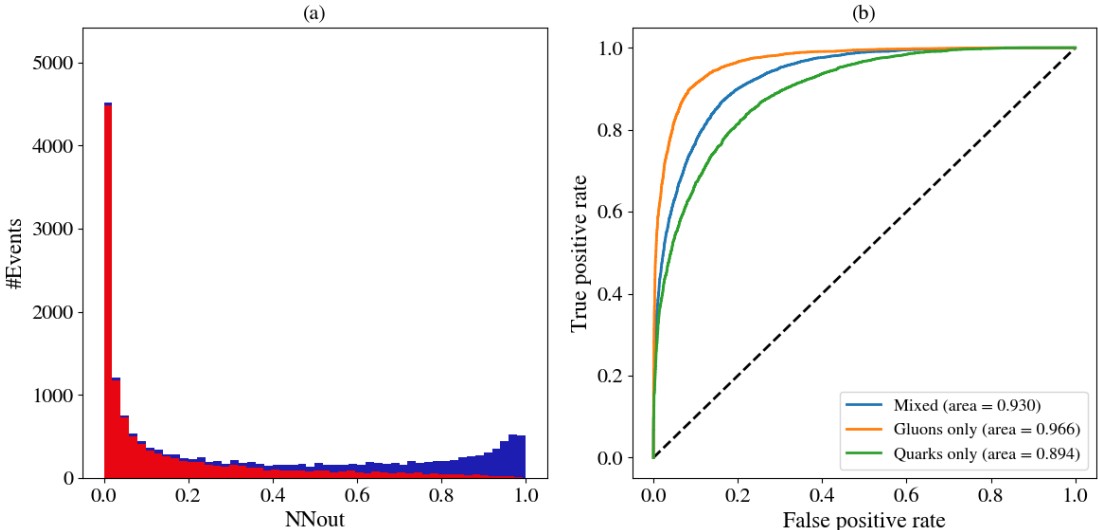

Figure 2: (a) Output of the neural network for signal (blue) and background (red). (b) ROC curves for the network.

GeV and $p_T > 50$ GeV. We notice a maximum drop of 0.009 for the gluon sample and 0.016 for the quark sample in the AuC value, showing that the effect of jet $p_T$ is limited.

We also investigate the stability of the network performance under variations of the simulation parameters. For this we applied the recommended variations [15] of the Pythia 8 parton shower and multiple parton interaction parameters based on the NNPDF23LO tune. These variations cover a range of data observables from ATLAS Run 1. Variation 1 is related to the underlying event activity, variation 2 is covering the jet shapes and substructure and the three variations 3 cover the effects of initial and final state radiation. As can be seen in table 4, the network performance is very stable for different tunes.

Table 4: Variations in the AuC for different Pythia tunes

| Parameter | +variation | -variation |
|---|---|---|
| Var1: UE activity | +0.001 | +0.005 |
| Var2: jet shapes and substructure | +0.005 | −0.006 |
| Var3a: ISR/FSR $t\bar{t}$ gap | +0.010 | +0.002 |
| Var3b: ISR/FSR 3/2 jet ratio | −0.003 | +0.008 |
| Var3c: ISR | +0.001 | +0.002 |

Finally we discuss the effect of concurrent proton-proton interactions or pile-up. Our set-up is not very well suited to evaluate its effects. The LHC experiments have been able to mitigate the effects of pile-up by using additional tracking and vertexing information, which is not available in DELPHES and our estimate is in this sense a worst case scenario. We simulate a sample with a pile-up of $\mu = 35$, on average 35 concurrent interactions, the average of the recent LHC run. The retrained network, without further optimization, shows a drop of 0.05 in the AuC. This shows that pile-up has a significant impact but that our method still works in the presence of pile-up. We would expect that some of the performance loss can be recovered in a full simulation and reconstruction chain.

# 6 Conclusion

We have presented an algorithm to identify jets from hadronic decay of charmonium states and have demonstrated that it works with a good efficiency of 36% for signal at a 100 times rejection of a background of quarks and gluons. Against a background of only gluons the algorithm works even better. The method also works for $J/\psi$ and heavier charmonium states and is relatively insensitive to the simulation parameters. The algorithm works in the presence of pile-up but at a significant loss of performance. This opens the possibility to use hadronic decay modes of charmonium in the search for rare decays to $c\bar{c}$ states that suffer from low statistics.

# Acknowledgments

NdG would like to thank the Duke University physics department for their hospitality.

**Funding information** SC would like to acknowledge support from the Duke REU Program through NSF Grant No. NSF-PHY-1757783.

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
