# Peer review of "Identifying hadronic charmonium decays in hadron colliders"

_SciPost Physics, doi:SciPost Phys. Core 2, 008 (2020)_

## Round 2 · Referee Report · Yuta Takahashi (Referee 1) · 2019-10-28

Strengths

The idea of tagging charmonium hadronic decays are quite interesting experimentally and unique attempt in this area. If this works as author mentions, there might be a nice gain in some of the physics analysis. (e.g. $H\to c\bar{c}$)

Weaknesses

I guess the performance of your tagger heavily depends on the $p_\text{T}$, and this argument is completely missing. Your study is based on the Z resonance, and thus performance is rather good, but there are many cases where people want to tag $J/\psi$ inside the B or D meson decays, where the performance heavily degrades (well, you are using the output of b-jet identification algorithm and thus will not be sensitive to inclusive B or D meson production, that have finite lifetime, but you can certainly remove this requirement).

Report

First of all, please run the spell-checker ... there are really a lot of typos, incomplete sentences; referee is not supposed to correct for your typo, but rather review the physics content. This is one of the minimum you should do before submitting the draft (as a manner). For typos, please see "Requested changes". Here, I mainly focus on the physics questions/comments.

Section2: - please clarify what 3S1, 3PJ stands for - Why gg2ccbar(3PJ)gm were not considered, like you did for $J/\psi$ and $\psi(2S)$? - Why qg2ccbar(3S1)q, qq2ccbar(3S1)g were not considered, like you did for $\chi_c$? - L7: "same sample" -> which sample? At this point, you haven't yet explained BG samples (you mention it in L8). Please rephrase. Also, How you get "gluon" jets from $Z^0\to q\bar{q}$ sample ?

Section-3: - L1: "tau-like" is not really true, as it also includes leptonically decaying taus. Please use more better phrasing. - L8: concerning to b-tag output. According to Table1, you seemed to use binary information, but which working point, how much the signal efficiency and fake rate? - 2nd paragraph: Please add explanation about How $\kappa$ and $\beta$ is determined. - Are these variables collinear safe?

section-4: 2nd paragraph: what do you mean by "gluon events"?

section-5: - First paragraph: the notation "100x" is better avoided. - L2: define "Auc" (Area under curve) as you are going to use it

Requested changes

Please find below what I spotted, but there might be more. Please double-check again your whole sentences.

Abstract: - L2: Here, you use non-italic for $J/\psi$, unlike you did in the text. I don't know the convention in abstract, but I feel more comfortable if you use the same font across whole paper. Please check. - L2: and algorithm -> an algorithm

Section 1: - page1, 3rd line from the bottom: uur -> our - page1, last line: overal -> overall

Section 2: - L4: extra space between qg2ccbar(3PJ)" and "q - L4: extra space between qq2ccbar(3PJ)" and "g - L5: "the process" does not have "minimum transverse momentum" -> rephrase it with more clear sentence - L5: "the process" does not have "invariant mass" -> rephrase it with more clear sentence - L2 from the bottom: $\Delta R$ is not defined.

Section 3: - L2: [5]. -> missing period - L4: you are missing ")" after $R_{track}$ - L6: you need to define "pt" - L12: ... demonstrated to be efficient in "the" distinguishing -> remove "the" - L15: use italic for "i" - L16: I would not use short-hand "w.r.t" - Figure 1 caption: parenthesis is not closed. Also, add unit. Please also add basic information such as the distribution is normalized to unity, and you are using 50% quark jets and 50% gluon jets ... The caption should be more self-explanatory. - Table 1, $R_{em}$: $\Delta_{R}$ -> $\Delta R$. Same for $R_{track}$

Section 4: - L6: as a measure "of" the separation power ... - 2nd paragraph: with and equal -> with "an" equal ... - Table 2: The caption requires more explanation. As outside readers (like me), none of the single item is understandable. You can also consider to put it to appendix.

Section 5: - L2: curve.s -> curves. - L5: extra space after "background rejection" - 2nd paragraph: "is" given in Table3 - Table 3, caption: Overview "of the" training results. Please also add more explanation to the caption what mixed means. - These variations cover "the a" range of ... -> please make it grammatically correct - Table4: Please add more explanation to the caption. For example, what is UE, what is ISR, FSR (you don't define them in the text), what is Var 3a, 3b.,3c etc.

Acknowledgement: I'm not sure if the abbreviation such as "NdG" and "SC" can be used in the paper: maybe the authors would like to ... ?

  • validity: good
  • significance: good
  • originality: good
  • clarity: good
  • formatting: acceptable
  • grammar: acceptable

Author:  Nicolo De Groot  on 2019-12-11  [id 674]

(in reply to Report 1 by Yuta Takahashi on 2019-10-28)
Category:
answer to question

Dear Referee,

thank you for the careful reading and your comments. We will release shortly an updated version that will address the requested changes and the comments of the other referee.

Regarding the mentioned weakness, the algorithm uses heavily the fact that charmonium jets are narrower and not surrounded by fragmentation and hadronization tracks, so it is not expected to work for charmonium decays inside b-jets. It would work at another momentum scale for charmonium jets coming from a color-singlet state.

Concerning the report, for section 2, 4 and 5 will be updated to clarify the points brought up here. The kappa and beta parameters are not optimized but a couple of discrete values are chosen, leading to different variables with discrimination power between signal and background. Variables with kappa > 0 are infrared safe. Those with kappa = 1 are collinear safe.

Regarding table 2, we would like to keep it in this form. For those using machine learning it contains relevant information on the network configuration.

---

## Round 2 · Referee Report · Anonymous (Referee 2) · 2019-11-6

Strengths

1- The paper demonstrates that J/Psi and other hadronic charmonia decays can be distinguished from quark and gluon jets under a number of assumptions, and quantifies the performance of an algorithm that does this. This can be useful input for phenomenological studies of potential analyses that consider these decays or serve as a template for such an algorithm to be implemented in an experiment.

2- While not without flaws as mentioned below, the writing is succinct and to the point and hence quite clear.

3- The algorithm itself seems reasonably solid.

Weaknesses

1- Motivation too vague. While the general motivation is clearly intriguing, it is not clear which performance of the identification algorithm is required for it to be useful in an actual analysis of the suggested H-> J/Psi gamma search. Without at least a rough study and given the large photon + jet production cross section compared to the Higgs production cross section and, in addition, a branching fraction of 3 x 10e-6, it remains unclear if the results of the presented study will be of any practical use.

2- Doubts on the performance: While the performance seems somewhat solid, a few doubts remain under realistic circumstances (in particular concurrent pp interactions) and of the assumptions made e.g. on the pT spectra.

3- Complexity of hadronic J/Psi decays not discussed and potentially not addressed: It is mentioned that J/Psi decays are similar to tau decays, but this is only partly true. A deeper discussion is missing.

4- Presentation: There are a number of obvious typos and grammatical mistakes.

Report

The presented study tries to distinguish hadronic charmonium decays from quark and gluon jets. An algorithm based on a number of higher-level input variables fed into a Deep Neural Network is discussed, similar to how tau lepton decays are identified in the ATLAS and CMS experiments. While a recent trend has been to include more low-level input variables to increase the performance of such taggers, the presented approach seems adequate for the purpose at hand given that the gain from using lower-level inputs is typically significant but far from an order of magnitude, and that similar approaches are in use by the ATLAS and CMS experiments and hence well understood. In the following, a list of general comments on the content is given. Comments on the language are included in the "Requested changes" section. It should be noted that, while the draft often reads smoothly, there is a potentially distractingly large number of grammatical mistakes and typographical errors.

Motivation: While the motivation to increase the acceptance for H-> J/Psi gamma makes sense on paper, it is a priori not clear which reconstruction and identification efficiencies are needed to make the channels with hadronic J/Psi decays competitive with the clean dimuon and dielectron final states. It is doubtful that a factor 100 rejection of jets is sufficient to reduce the background to acceptable levels. A more thorough investigation of the H->J/Psi gamma search or at least a back-of-the-envelope calculation would strengthen the motivation for this paper. Alternatively, the motivation could be expanded to other processes, with a similar demonstration of the potential usefulness as requested for the H->J/Psi gamma decay.

Pileup: With the search for H->J/Psi gamma in mind, which I assume is to be performed at the High-Luminosity LHC, the impact of pileup (concurrent pp interactions) is very relevant for the identification of hadronic charmonium decays, as it is for the reconstruction and identification of tau decays. Can an estimate of the possible impact of pileup be given?

J/Psi is not like a tau, since its decay most often does not involve hadronic resonances, whereas the tau either decays to a single charged particle and neutrino or via hadronic resonances in the vast majority of the overall decays (See PDG 2018 particle listing on J/Psi http://pdg.lbl.gov/2018/listings/rpp2018-list-J-psi-1S.pdf, or https://www.sciencedirect.com/science/article/pii/0370157389900744): - Less than 2% of decays involve hadronic resonances - Around 13% into stable hadrons - To be contrasted to around 88% total decays to hadrons - On the other hand, J/Psi does typically not have invisible decay products - This should be elaborated on in more detail; one could even make use of the information from the preferred decay modes, as done for tau decays.

pT and eta spectra: - It should be demonstrated explicitly that the pT and eta spectra for the signal charmonium decays and for quark and gluon jets (separately) are identical or sufficiently similar. Ideally, they should be made identical (Section 2 mentions "similar", but this is a too vague statement: How similar?) - In addition to the eta distributions having to be identical, the upper cutoff should also be introduced.

Size of simulated event samples: - Given that the tau taggers in ATLAS https://cds.cern.ch/record/2688062/files/ATL-PHYS-PUB-2019-033.pdf and CMS https://arxiv.org/abs/1809.02816 are trained with millions of simulated events, the 17 k events used for the training seem to be at the low end. An investigation of the effect of increasing the sample size would be useful to understand whether a potentially large gain in performance is still possible with the method at hand. The other parameters of the deep neural network and the optimisation procedure seem adequate.

Requested changes

1- Motivate which performance of the tagger is required to make its inclusion in the H-> J/Psi gamma search useful, or alternatively find additional motivation.

2- Discuss and investigate the impact of concurrent pp interactions.

3- Elaborate on and possibly investigate in more detail the various hadronic J/Psi decays.

4- Make pT/eta spectra of signal and background identical or show that these are indeed sufficiently similar.

5- Address doubts on the size of the event samples being sufficient.

6- Improve figure 1 and discuss questions below in the text: - Figure 1 caption, 2nd sentence: The J/Psi signal distributions are shown in blue, whereas the background from quark and gluon jets is shown in orange. - Figure 1: It seems like the distributions are normalised to the same area, but it would be better to clarify in the caption. - Figure 1: There seem to be two distributions of J/Psis in DeltaEta and DeltaPhi. It would be interesting to understand how they arise. - Figure 1: Are there any structures in the jet mass that are exploited by the network that are invisible because of the coarse binning? - Figure 1: What leads to the spikes in the n_ch distribution (in particular around 8 for quark and gluon jets)

7- Improve language, run spellchecker, and avoid grammatical mistakes, in particular the following list: General - Run spellchecker - Various full stops omitted after citations (e.g. end of section 2)

Abstract: - we present and algorithm -> we present an algorithm

Introduction

  • uur ->our
  • overal -> overall

Observables - in the distinguishing -> in distinguishing - a background samples -> a background sample

Machine Learning - with and equal amount -> with an equal amount

Results - curve.s -> curves - as against -> than against

8- Citations: Cite Keras according to their preference https://keras.io/getting-started/faq/#how-should-i-cite-keras

  • validity: ok
  • significance: ok
  • originality: good
  • clarity: good
  • formatting: reasonable
  • grammar: acceptable

Author:  Nicolo De Groot  on 2020-02-14  [id 736]

(in reply to Report 2 on 2019-11-06)
Category:
answer to question

Dear referee, thank you for the careful reading and useful comments, we will fix the language issues and below reply to your questions and remarks.

1- Motivate which performance of the tagger is required to make its inclusion in the H-> J/Psi gamma search useful, or alternatively find additional motivation.

We would argue that this paper is the first one to demonstrate out the possibility of identifying hadronic charmonium decays and this makes it interesting enough by itself to be published regardless of the numerical impact on the H-> J/Psi gamma search. A realistic evaluation of this impact would be a research project on its own. While this method will suffer from a significantly larger background than the clean J/Psi -> 2mu decay, it will also have larger statistics. It will target 88% of the hadronic J/Psi decays rather than 6% of the muonic decay. In addition it targets 4 other charmonium resonances, leading to a data sample which is about 40x larger.

Looking at complicated LHC analyses like the ttH search we see that many channels with different purity, efficiency and significance are combined for the final result. Also we note that analyses channels are further optimized over time, and channels which are initially not included are contributing at a later stage. While we do not expect to “beat” the 2mu channel, we do expect to make a contribution.

An obvious other process where this method could be of relevance is the search of the rare decay Z->J/Psi gamma. The method of identifying hadronic decays from color singlet hadrons can easily be extended to other mesons, like Ds() for the decay W-> Ds() gamma. Finally this paper is discussing charmonium production in hadron colliders, but the method would also work well in e+e- colliders like ILC or CLIC where the jet background is substantially lower.

2- Discuss and investigate the impact of concurrent pp interactions.

It is not possible to fully address this issue within the DELPHES simulation framework, since a number of tools to suppress the effects of pile-up are not available. We see that pile-up has initially affected identification of leptons and heavy flavor jets, but that experiments have improved there selection over time to make them more robust against pile-up. We would hope and expect that this is also the case for this algorithm. As a check we investigated the performance using samples with a pile-up of 35 concurrent interactions, the average for the LHC run 2. After retraining, but no further optimization, we observe a significant drop of 0.050 in the AuC. This does show that our method still works in presence of pile-up and we would hope to recover some of the performance in the optimization. See attached picture.

3- Elaborate on and possibly investigate in more detail the various hadronic J/Psi decays. The various hadronic decay modes of the charmonium modes we are investigating have in common that unlike in the case of tau decay, where a few final states dominate the decays, there are no dominant resonances. This makes charmonium more suitable for an inclusive approach using machine learning as presented in this paper.

4- Make pT/eta spectra of signal and background identical or show that these are indeed sufficiently similar.

At generator level the spectra for charmonium and the gluon are the same since they come from the same 2->2 process. The quark spectrum is a bit more spread out but peaks at a similar value. After reconstruction there is a difference in pt spectrum. The charmonium jets have a higher average pt than the gluon and quarks jets. For the quark jets one of the reasons are the semi-leptonic decays of the b and c quarks jets where the neutrinos take away energy. For gluon jets, the reason is a bit more subtle. Gluon jets are wider and at 30-50 GeV this means that not all the jet constituents end up in a 0.4 cone and the reconstructed energy is lower.

Since some of the input variables have a pt dependence this could translate into fake separation power. We check the effect of the difference in momentum distributions after reconstruction by splitting our test sample in 3 bins: jet pt < 35 GeV, 35GeV < pt < 50 GeV and pt > 50 GeV. We see a maximum drop of up to 0.010 in the AoC for gluons versus signal and 0.018 for quarks versus signal and conclude that the network performance does not depend strongly on the pt of the jets.

We do not observe any significant differences in eta spectrum. We will mention this check in the new version of the paper.

5- Address doubts on the size of the event samples being sufficient.

Now this is interesting. When we optimized the neural network we also played with sample sizes and our conclusions indicated that our current sample was large enough. When we repeat the training with a sample which is 4x larger we see a small but reproducible improvement of the AoC of about 0.003. We will update the paper with the results from the larger sample.

6- Improve figure 1 and discuss questions below in the text: - Figure 1: There seem to be two distributions of J/Psis in DeltaEta and DeltaPhi. It would be interesting to understand how they arise. The first peak seems to be correlated with the presence of a K0

  • Figure 1: Are there any structures in the jet mass that are exploited by the network that are invisible because of the coarse binning? There are peaks around the charmonium masses in the track mass for decays without neutrals and in the jet mass for all decays that are perfectly reconstructed. Since both are input variables to the neural network this is taken into account
  • Figure 1: What leads to the spikes in the n_ch distribution (in particular around 8 for quark and gluon jets) This is a binning problem with two charged multiplicities ending up in the same bin. It will be fixed in the update.

Attachment:

---

## Round 2 · Referee Report · Anonymous (Referee 3) · 2019-12-11

Report

Hello,

This is a very interesting paper with a very well-motivated and important aim. I am putting some general questions and comments first, followed by a list of some minor editorial issues.

General questions and comments:

o 1 Introduction:
-Please add some theory references relevant to the motivation for H to J/Psi\gamma searches (e.g., papers by G. Perez, Y. Soreq, E. Stamou, and K. Tobioka).

-Please also add some recent experimental references relevant to the H to charm searches

-Of course there are also important SUSY searches at hadron colliders which would also benefit from charm tagging.

-You don't mention or reference the charm taggers that have already been developed and used by others (e.g., ATLAS and CMS).

o 2 Simulated Samples:
-What kind of hadron collider is being considered here and what was the CoM for the simulated samples?

-What has been done to model in-time pile-up? Has any been included in the samples at all? Since these H to charm searches will most certainly continue to be relevant well into the HL-LHC era, might want to investigate the impact with a mu~200 scenario.

-What are the statistics of the simulated samples? This info appears in the next section but perhaps it is better placed here?

o 3 Observables:
-Which b-jet identification algorithm are you referring to here? Which benchmark (e.g., 'loose', 'tight', etc., would be helpful to comment on the performance of this b-tagging algorithm on a sample like ttbar)?

-Regarding the description of your variables, Lower mass, lower multiplicity and narrower than what? You don't say.

-Which b-tagging algorithm is used (e.g. mentioned in Table 1)? What is the selection that is used? Is it tight or loose, for example... need to give a benchmark, wrt fake rate and ID efficiency on something like ttbar.

-Table 1: Do you have a reference for the Les Houches Angularity variable (e.g., shown in Table 1)? Also, what are these lambdas (e.g., for the width and mass)?

-Figure 1: Exactly which quarks are you considering here?

o 4 Machine Learning:
-What is a 'hyperparameter'? Why not simply 'parameters of the neural network'?

-Is the Architecture designation in Table 2 standard? I could make a guess regarding what I think you mean, but probably best to state it explicitly for the sake of clarity.

-The learning and dropout rates are in units of what?

o 5 Results:
-Abbreviations like AuC and ROC are almost jargon, can you please write these out in the first instance?

-Which variables are the most powerful? Which are not?

-Figure 2 right-most plot and in the body of the text: exactly which quarks?

-Table 4: Perhaps I'm just not understanding what this is, but is the Var3a (ttbar gap) relevant here?

-Along those lines: did you consider investigating the fake tagging rate on b-jet rich or ttbar rich samples?

o 6 Conclusion:
-You're quite vague regarding the rare decays mentioned on the last line here.

-How does the performance of the tagger you developed compare with the charm taggers that have already been developed and used by others (e.g., ATLAS and CMS)?

Minor editorial comments which require changes:

o Title and authors:
-Comma after the second author is not needed

-Steet --> Street

-Not clear why the asterisk appears before the first author's email address

o Abstract:
-present and --> present an

-I would suggest that you still include the 'c' in the subscripts that denote the chi c1 and c2 states

o 1 Introduction:
-The grammar in the last line of the second paragraph here needs to be corrected

-What is a neutral tau? I understand what you mean, but would suggest phrasing it differently.

-uur --> our

o 2 Simulated Samples:
-100cm --> 100 cm

-I think that Delphes is usually written all-caps

-anti-k_t (no capitalization and write the 't' as a subscript)

-Full-stop at the end of the last sentence.

o 3 Observables:
-You're missing a full-stop after reference [5].

-I think that it would be helpful to explain what some of these variables are rather than leaving the reader to understand these via reference 5.

-The grammar in the first line of the second paragraph here needs to be corrected.

-w.r.t --> You should write this out here and not use an abbreviation (both in the main text and in Table 1)

-Table 1: The first reference to this table really belongs closer to the start of Section 3 where you first mention the variables. It is better being placed there, so that the reader can then immediately have a look and understand what each of these are.

-Also Table 1: In general, I think that it is better to write "jet candidate". Perhaps a clarification in the caption would suffice.

-You're using two different conventions for writing transverse momentum here. Would suggest using the one with a capital 'T' throughout the entire document, including all tables.

o 4 Machine Learning
-The grammar in the last line of the first paragraph, as well as the second line of the second paragraph, needs to be corrected.

o 5 Results
-curve.s --> curves

-I would suggest that you say 'factor of 100' rather than 100x. The latter seems too much like slang.

o 6 Conclusion
-The grammar in the first line of the first paragraph needs to be corrected.

o Figure 1:
-The legend also needs to be in the plots. Also, although the article is online and in color, I would really suggest using solid and dashed lines in case this is printed out in B&W.

-You need to close the parenthesis in the figure caption.

o Figure 2:
-Left plot: I know it seems obvious, but for the sake of the more novice reader you should include a legend to indicate which is signal and which is background.

-Right plot: I'm not sure why you need the dashed line here. Again, would be good if one is still able to distinguish the lines when printed in B&W.

-Caption: Add a full-stop at the end.

o References:
-The arXiv hyperlink for Ref. 7 appears to be broken
  • validity: -
  • significance: -
  • originality: -
  • clarity: -
  • formatting: -
  • grammar: -

Author:  Nicolo De Groot  on 2020-03-03  [id 751]

(in reply to Report 3 on 2019-12-11)

Thanks you for your comments, we will address most in our updated version, for the remaining questions and remarks, please see below:

o 1 Introduction:
-Of course there are also important SUSY searches at hadron colliders which would also benefit from charm tagging.
Yes but only if there are charmonium mesons in the final state.
-You don't mention or reference the charm taggers that have already been developed and used by others (e.g., ATLAS and CMS).
This is not so relevant since the are tagging really different things, open charm vs charmonium.

o 2 Simulated Samples:
-What has been done to model in-time pile-up? Has any been included in the samples at all? Since these H to charm searches will most certainly continue to be relevant well into the HL-LHC era, might want to investigate the impact with a mu~200 scenario.
The baseline samples are without pile-up. In the updated paper we discuss results for a pile-up of 35, the average for Run2.

o 3 Observables:
-Which b-jet identification algorithm are you referring to here? Which benchmark (e.g., 'loose', 'tight', etc., would be helpful to comment on the performance of this b-tagging algorithm on a sample like ttbar)?
This is the standard ATLAS b-tagging algorithm. The new text will specify its benchmark performance.
-Regarding the description of your variables, Lower mass, lower multiplicity and narrower than what? You don't say.
Than quark and gluon background
-Table 1: Do you have a reference for the Les Houches Angularity variable (e.g., shown in Table 1)? Also, what are these lambdas (e.g., for the width and mass)?
[reference 7]
-Figure 1: Exactly which quarks are you considering here?
All but top, the quarks from the sample described in section 2.

o 4 Machine Learning:
-What is a 'hyperparameter'? Why not simply 'parameters of the neural network'?
In machine learning a hyperparameter are used for parameters which have been used during the training phase, but not the final weights at the end of the training.
-Is the Architecture designation in Table 2 standard? I could make a guess regarding what I think you mean, but probably best to state it explicitly for the sake of clarity.
Yes it is a fairly standard architecture.
-The learning and dropout rates are in units of what?
These are dimensionless numbers

o 5 Results:
-Which variables are the most powerful? Which are not?
The width and mass related variables are the most powerful. Given their correlation it would be misleading to pick out a single most powerful one. The jet charge does not seem to bring much.
-Figure 2 right-most plot and in the body of the text: exactly which quarks?
The quarks from the Z-> qq background sample described in section 2
-Table 4: Perhaps I'm just not understanding what this is, but is the Var3a (ttbar gap) relevant here?
These are a set of recommended variation on the Pythia parameters which give a good coverage of the systematic uncertainties related to parton shower and multiple interaction parameters. The Var3a is part of this set.
-Along those lines: did you consider investigating the fake tagging rate on b-jet rich or ttbar rich samples?
No, light quarks are more prevalent at the LHC. Also the b-tagging actually rejects b-jets and b jets are present at 21% in the Z->qq sample.

o 6 Conclusion:
-How does the performance of the tagger you developed compare with the charm taggers that have already been developed and used by others (e.g., ATLAS and CMS)?
Again this is not so relevant. The existing charm taggers are using decay length information. Since charmonium has zero decay length these taggers are very inefficient for charmonium mesons. To our knowledge this is the first inclusive charmonium tagger and jets from open charm are considered a background.

---

## Round 3 · Referee Report · Yuta Takahashi · 2020-3-19

Strengths

Same as previous comment.

Weaknesses

Same as previous comment.

Report

The updated version has significant improvement compared to the previous version. At this point, I have just a few more comments.

1) Table1:
"abs(qj): pT weighted charge sum, kappa = 0.5" -> Kappa seems not defined in the text (it is different kappa as Eq.1 right?). Can you add description what it means or rephrase?

2) Figure1:
Can you add y-axis scales (e.g. 0, 0.5, 1 or something like that)? It is possible to roughly draw this information by eye but it is difficult for the variables with large number of binning.

3) Figure2, left plot:
I guess the plot is "stacked" (not overlaid). Can you be explicit about it?

Left over from last comment
- Why qg2ccbar(3S1)q, qq2ccbar(3S1)g were not considered, like you did for $\chi_c$?

  • validity: good
  • significance: good
  • originality: good
  • clarity: good
  • formatting: good
  • grammar: good

Author:  Nicolo De Groot  on 2020-03-31  [id 785]

(in reply to Report 1 by Yuta Takahashi on 2020-03-19)
Category:
answer to question

Dear Dr. Takahashi,

thank you for your comments. They seem quite minor and we hope to address them in the final version.

1) Table1: "abs(qj): pT weighted charge sum, kappa = 0.5" -> Kappa seems not defined in the text (it is different kappa as Eq.1 right?). Can you add description what it means or rephrase?

Will be fixed in the final version

2) Figure1: Can you add y-axis scales (e.g. 0, 0.5, 1 or something like that)? It is possible to roughly draw this information by eye but it is difficult for the variables with large number of binning.

We will try to see if this is an improvement. The plot is already very crowded with 20 subplots.

3) Figure2, left plot: I guess the plot is "stacked" (not overlaid). Can you be explicit about it?

Will be fixed in the final version

Left over from last comment - Why qg2ccbar(3S1)q, qq2ccbar(3S1)g were not considered, like you did for χ_c?

The reason for it is quite trivial. There is no qg2ccbar(3S1)q switch in Pythia8 for color singlet production. See http://home.thep.lu.se/~torbjorn/pythia82html/OniaProcesses.html

---

## Round 3 · Referee Report · Anonymous · 2020-3-30

Strengths

1- The paper demonstrates that J/Psi and other hadronic charmonia decays can be distinguished from quark and gluon jets under a number of assumptions, and quantifies the performance of an algorithm that does this. This can be useful input for phenomenological studies of potential analyses that consider these decays or serve as a template for such an algorithm to be implemented in an experiment.

Weaknesses

1- Vague motivation. While the general motivation is intriguing, it is not clear which performance of the identification algorithm is required for it to be useful in an actual analysis of the suggested H-> J/Psi gamma search. Without at least a rough study and given the large photon + jet production cross section compared to the Higgs production cross section and, in addition, a branching fraction of 3 x 10e-6, it remains unclear if the results of the presented study will be of any practical use.

Report

The updated version addresses a few minor doubts on the performance, in particular in the presence of pileup, the size of the training samples, and the impact of the used pT distributions on the performance. I have no reason to doubt that the obtained results are solid.

With that said, a much stronger paper could have been obtained by demonstrating that the performance of the algorithm is good enough for a concrete analysis to become competitive. Without this demonstration, it remains to be seen if the performance of the presented algorithm is sufficient to be practically useful.

The presented study will however allow for such studies to be performed, and hence be potentially useful for experimentalists or phenomenologists who try to explore hadronic charmonium decays. To indeed be useful, the obtained ROCs should be made available in machine-readable format, and it would be good to also give the signal efficiencies for a lower background efficiency, as suggested below.

Requested changes

1- It would be instructive to not only give signal efficiencies for jet misidentification efficiencies of 1 per cent, but also for a much smaller one, e.g. 1 per mille. These cannot be inferred from the ROCs since they are not shown with logarithmic misidentification efficiency axis, but will likely be important in practical applications, similar to tau identification. In addition, the AUCs are not very instructive measures of the performance.

2- Abstract & Conclusion: Both the abstract and the conclusion should mention the performance of the algorithm.

3- There are still a few minor language and style issues:
- Caption figure 1, "used to" -> "used for"; Distribution -> Distributions
- Add reference for dropout

---

## Round 3 · Referee Report · Anonymous · 2020-4-24

Report

Thanks a lot for addressing my comments and questions on the earlier version of your draft. I'm satisfied with the answers, as well as your changes. I only have a few very minor additional suggestions:

o General comments:
-The PDG and most papers seem to use lowercase \psi when referring to the J/\psi particle. I'm not sure why you've decided to use \Psi. Similar comment for \psi(2s).

-You're using both p_T and p_t in the text, tables and figures. I would suggest using p_T consistently throughout the document.

o Section 5:
-Write out "Figure 2" when you start the paragraph.

-On Page 5, where you list the ranges of the three pT bins, I'd recommend adding a space between the numbers and "GeV" units.

o References Section:
-Ref. 2: The actual paper title uses a capital 'H'.

-Ref. 4: Something is really wrong with the paper title for this reference. Looks duplicated and one symbol is not recognized as a '?' appears instead.

-Ref. 10: I don't think that the capitalization used here matches that in the actual paper title.

o Figure 1: Please add a full stop at the end of the caption.

---

## Round 3 · Author Response

New version with comments of the referees addressed

---

## Round 3 · List of Changes

1. Cross-check with pile-up implemented
2. Higher statistics used in training
3. Cross-check for pt dependence inplemented
4. Improved binning in plots
5. References on rare Higgs decay added
6. Textual improvements, typos

---

## Round 4 · Author Response

Version correcting referees comments

---

## Round 4 · List of Changes

- efficiency at background rejection of 1000 quoted
- reference to dropout added
- few typos corrected

Resubmission 1910.05334v3 on 17 March 2020
Submission 1910.05334v2 on 22 October 2019

---

## Editorial Decision

published